# Temporal profiles of avalanches on networks

James P. Gleeson [1] & Rick Durrett[2]

An avalanche or cascade occurs when one event causes one or more subsequent events, which in turn may cause further events in a chain reaction. Avalanching dynamics are studied in many disciplines, with a recent focus on average avalanche shapes, i.e., the temporal profiles of avalanches of fixed duration. At the critical point of the dynamics, the rescaled average avalanche shapes for different durations collapse onto a single universal curve. We apply Markov branching process theory to derive an equation governing the average avalanche shape for cascade dynamics on networks. Analysis of the equation at criticality demonstrates that nonsymmetric average avalanche shapes (as observed in some experiments) occur for certain combinations of dynamics and network topology. We give examples using numerical simulations of models for information spreading, neural dynamics, and behavior adoption and we propose simple experimental tests to quantify whether cascading systems are in the critical state.

[1] MACSI, Department of Mathematics and Statistics, University of Limerick, Limerick, Ireland. [2] Department of Mathematics, Duke University, Durham, NC 27708, USA. Correspondence and requests for materials should be addressed to J.P.G. (email: james.gleeson@ul.ie)

The dynamics of avalanches or cascades are studied in many disciplines. Examples include the spreading of disease (or information) from human to human[1, 2], avalanches of neuron firings in the brain[3], and the "crackling noise" exhibited by earthquakes and magnetic materials[4]. Of particular interest are cases with dynamics poised at a critical point, where universal scalings of avalanches are observed. The most commonly studied feature of such systems is the distribution of avalanche sizes, which has a power-law scaling at the critical point. The observation of heavy-tailed distributions of avalanche sizes has therefore been used to indicate whether a system is critical. However, power-law distributions can also arise from mechanisms other than criticality[5, 6], so recently attention has focussed more upon the temporal aspects of avalanches, which also exhibit universal characteristics at criticality.

The average avalanche shape is determined by averaging the temporal profiles of all avalanches that have a fixed duration $T$. At criticality, the average avalanche shape is a universal function of the rescaled time $t/T$, meaning that the average avalanche shapes for different durations can be rescaled to collapse onto a single curve[4]. This feature has recently been used as a sensitive test for criticality in a range of dynamics, from the Barkhausen effect in ferromagnetic materials[7] to neural avalanches[3, 8] and electro-encephalography recordings from hypoxic neonatal cortex[9]. While average avalanche shapes are typically symmetric (e.g., parabolic) functions of time, nonsymmetric (left-skewed) avalanche shapes have also been observed in experiments. For example, early observations of nonsymmetric avalanche shapes in experiments on Barkhausen noise[4] raised doubts about whether the theoretical model used in refs. [10, 11] was in the correct universality class. Although this discrepancy between theory and experiment was later resolved by a more detailed theory for avalanche propagation[12, 13], several instances of non-symmetric avalanche shapes (e.g., the neural avalanches in ref. [3]) still lack explanation. Despite some progress in modeling avalanche profiles using random walks[9, 14] and self-organized criticality models[15–18], the factors that cause nonsymmetric average avalanche shapes remain poorly understood.

The characteristics of avalanches that occur on networks depend on both the network connectivity and the node-to-node dynamics of the cascade[19]. Cascading models have been applied, for example, to power-grid blackouts[20], epidemic outbreaks[21], and to the propagation of memes (pieces of digital information) through online social networks[22, 23]. The distribution of avalanche sizes at criticality is known to depend non-trivially on the degree distribution of the underlying network[24], but the time dependence of cascades has not been studied from this perspective.

In this paper, we focus on the temporal profile of cascades, i.e., the average avalanche shape, and how it is affected by the network degree distribution. Using a mathematical derivation of the average avalanche shape for Markovian dynamics (in both critical and noncritical cases), we demonstrate that—as in other universality-breaking examples[25]—networks with heavy-tailed degree distributions can give rise to qualitatively different results from those found on networks with finite-variance degrees. However, the dynamics of the avalanching process are also important: we show that in fact it is the interaction between the dynamics and the network topology that determines whether average avalanche shapes are symmetric or not.

## Results

**Average avalanche shapes.** To define the average avalanche shape, we consider the set $S_T$ of all avalanches that are of duration $T$ (meaning that the avalanche has terminated at a time $T$ after its

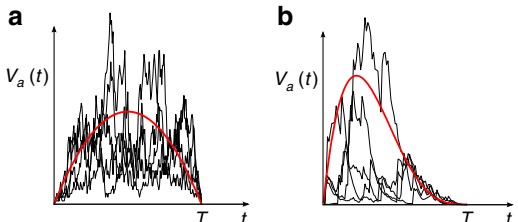

**Fig. 1** Examples of average avalanche shapes. In each panel, the black curves show five examples of individual avalanches that all have duration $T$. The average avalanche shape for the duration $T$ (red curve) is found by averaging the temporal profiles of all such avalanches. Typically, the average avalanche shape is symmetric (e.g., parabolic) as in **a**, but nonsymmetric avalanche shapes like **b** have also been observed (e.g., Fig. S4 of ref. [3])

first event, with no further events occurring at any time $t > T$). Each avalanche $a$ in the set $S_T$ is described by a function $V_a(t)$, which is the number of events that occur at time $t$ in that avalanche (so $V_a(t) = 0$ for $t > T$). The avalanche shape for the duration $T$ is defined as the average of the functions $V_a(t)$, taken over all the avalanches $a$ in set $S_T$, see Fig. 1.

**Dynamics and networks.** We consider networks that are defined by their degree distributions, but are otherwise maximally random ("configuration-model" networks[26]). For undirected networks, the degree distribution $p_k$ is the probability that a randomly chosen node has degree (number of neighbors) equal to $k$. For directed networks, the joint degree distribution $p_{jk}$ is the probability that a random node has in-degree $j$ and out-degree $k$. We denote the mean degree by $z$ (so $z = \sum_k k p_k$ for undirected networks and $z = \sum_{j,k} k p_{jk} = \sum_{j,k} j p_{jk}$ for directed networks). Such configuration-model networks are locally tree-like, which facilitates the use of the branching process approximations that we employ. We assume that the networks consist of a single connected component (a strongly connected component in the case of a directed network[26]) and that they are large enough to permit us to use infinite-size approximations.

Our focus is on discrete-state dynamics, where each node of the network can be in one of a set of discrete states at each moment in time; transitions between states may occur continuously in time, or only at discrete-time steps. Cascades occur when nodes successively switch to one specific state, which we will call the "active" state; we will generically refer to all other states as "inactive." Once a node is activated (i.e., once it transitions to the active state), it affects its neighboring nodes by increasing the probability that they will also become activated at a later time. We focus on unidirectional dynamics, meaning that in the case where a node activates some of its neighboring nodes and then subsequently becomes inactive, the neighboring nodes cannot directly reactivate it. One important class of such dynamics includes cases where an activated node cannot subsequently return to the inactive state, and so cannot be reactivated (this class is called "monotonic dynamics" in ref. [27]). Another class takes place on tree-like-directed networks, which have negligible numbers of loops, so that activation of node $i$ can affect its out-neighbors, but there exists no path for the out-neighbors to subsequently affect the state of node $i$ (even if node $i$ returns to the inactive state). Each cascade is assumed to be initiated by a randomly chosen single node, called the "seed" node, which is activated at the beginning of the process while all other nodes are inactive; subsequent to the activation of the seed node, the cascade of activation of nodes proceeds according to the rules of the model under consideration.

One example of such dynamics is given by threshold models. In a threshold model, each node $i$ of an undirected network possesses a positive threshold $R_i$ that is assigned randomly from a distribution. When an inactive node $i$ of degree $k_i$ is chosen for updating it considers the number $m_i$ of its neighbors who are active, and makes a decision according to the rules of the specific model. In the Watts threshold model[28], for example, node $i$ becomes active if the fraction $m_i/k_i$ is greater than, or equal to, the node's threshold $R_i$, i.e., the node activates if the fraction of its neighbors who are active is sufficently large, otherwise the node remains inactive. An alternative type of threshold model is that of Centola and Macy[29], wherein node $i$ activates if the total number of active neighbors (rather than the fraction of such neighbors) is large enough: $m_i \geq R_i$ (see also the discussion of threshold models and their relation to coordination games in ref. [27]).

Cascades also occur in the neuronal dynamics model of ref. [3], where each node in a weighted, directed network represents a neuron. The weight $\phi_{ij}$ on the edge connecting node $i$ to node $j$ is chosen at random from a uniform distribution on the interval $(0, \phi_{max})$, where $\phi_{max}$ is a tunable parameter[30, 31]. (In ref. [3] the values of $\phi_{ij}$ are instead inferred from experiments on neural networks). Using the same model parameters as ref. [3], the neurons are modeled by binary-state elements as a very simple approximation of integrate-and-fire dynamics: whenever neuron $i$ fires (becomes active), it causes neuron $j$ to become active (in the next discrete-time step) with probability $\phi_{ij}$. After a neuron fires, it is returned to the inactive state in the next time step. In ref. [3], exogenous input noise is added to the system to ensure continuous neural activity. Since we focus purely on the avalanche dynamics, we instead randomly select a node to be the seed node of the cascade and activate it in the first time step, and record the ensuing avalanche of activations.

A final example of cascade dynamics is given by the model of ref. [32] for meme propagation on a directed social network (like Twitter). In this model, each node (of $N$) in a directed network represents a user of the social network. The out-degree $k_i$ of a node $i$ is the number of its "followers" in the network: these are the users that receive the "tweets" (or distinct pieces of digital information, generically called "memes") sent by node $i$. Each user also retains a memory of the last meme received from the nodes it follows via a "screen" that is overwritten when a new meme is received. (More realistic models that incorporate longer memory are described in refs. [33, 34]). In each time step (with $\Delta t = 1/N$), one node is chosen at random and with probability $\mu$ the node "innovates" by creating a new meme, placing it on its screen, and tweeting this meme to all its followers (where the new meme overwrites any existing memes on their screens). Alternatively (with probability $1 - \mu$), the chosen node "retweets" the meme that is currently on its screen. A newly innovated meme can therefore experience an avalanche of popularity as it is retweeted multiple times before it eventually is forgotten by all users in the network, at which time the avalanche terminates. The analyses of refs. [32] and [34] show that the avalanche dynamics of the memes are critical in the limit $\mu \to 0$ and subcritical for $\mu > 0$.

Other examples of unidirectional dynamics to which our results apply include the zero-temperature random-field Ising model[4] and susceptible-infected-recovered disease-spread models with fixed recovery times[21], such as the independent cascade model for information diffusion[35].

**The offspring distribution.** The central quantity in our branching process analysis is called the offspring distribution, denoted by $q_k$ ($k = 0, 1, 2, \ldots$). Roughly speaking, this distribution gives the likelihood that if a cascade of activation reaches a node (i.e., if one of the node's neighbors becomes active), that the node

will activate and expose $k$ other neighboring nodes to potential activation (see Supplementary Note 1 for details of the branching process approximation). For an undirected network, the offspring distribution $q_k$ can usefully be expressed in terms of a simple degree-dependent quantity $\hat{q}_k$ that is defined as:

$$\hat{q}_k^{(undir)} = \frac{k+1}{z} p_{k+1} v_{k+1}, \qquad (1)$$

noting that the probability of reaching a node of degree $k + 1$ by traveling along a random edge is $\frac{k+1}{z} p_{k+1}$, and if this edge spreads the activation to the node it has $k$ remaining inactive neighbors. The quantity $v_k$ is the probability that a node $i$ of degree $k$ is vulnerable[28], meaning that the activation of a single neighboring node (at time $t_1$) will lead to the activation of node $i$ at some time $t > t_1$, assuming that no other neighbor of node $i$ becomes active by time $t$. Note that the sum $r = \sum_{k=0}^{\infty} \hat{q}_k \leq 1$ is the probability that a node reached by traveling along a random edge is vulnerable. The relationship between $\hat{q}_k$ and the offspring distribution $q_k$ is given by Eq. (5) below, but our main qualitative results depend only upon the large-$k$ scaling of $\hat{q}_k$ (see Supplementary Note 1).

For a directed network, the definition of $\hat{q}_k$ is

$$\hat{q}_k^{(dir)} = \sum_j \frac{j}{z} p_{jk} v_{jk}, \qquad (2)$$

where the factor $\frac{j}{z} p_{jk}$ represents the probability of reaching a node of in-degree $j$ and out-degree $k$ by traveling along a random edge of the network. The vulnerability $v_{jk}$ is the probability that the activation of a single in-neighbor (at time $t_1$) of a node $i$ (of in-degree $j$ and out-degree $k$) will lead to the activation of node $i$ at some time $t > t_1$, assuming no other in-neighbor of node $i$ becomes active by time $t$. The probability $r$ is defined as for the undirected case, and the relationship between $\hat{q}_k$ and the offspring distribution is again given by Eq. (5) below.

The branching number defines whether the dynamical process on a given network is subcritical, critical, or supercritical. The branching number is the mean of the offspring distribution, i.e., the expected number of "children" per "parent", and it can be expressed as:

$$\xi = \sum_k k q_k = \sum_k k \hat{q}_k, \qquad (3)$$

see Supplementary Note 1. The value $\xi = 1$ is the critical value, separating the subcritical case ($\xi < 1$) from the supercritical case ($\xi > 1$). In the critical case, power-law distributions of avalanche sizes are observed[32, 34] but in this paper we, focus on the temporal profiles of the avalanches.

**Average avalanche shape.** The detailed derivation of our results from the theory of Markov branching processes is given in Supplementary Note 2. For models with continuous-time updating, a particularly simple result is found: the average avalanche shape $A(t)$ for avalanches of duration $T$ can be expressed as:

$$A(t) = \frac{Q(T-t)[f'(Q(T)) - f'(Q(T-t))]}{f(Q(T-t)) - Q(T-t)}, \qquad (4)$$

where $f(s)$ is the generating function for the offspring distribution,

$$f(s) = \sum_k q_k s^k = \frac{1}{r} \sum_k \hat{q}_k (1 - r + rs)^k \qquad (5)$$

and $Q(t)$ is the fraction of avalanches that are extinct by time $t$, which is given by the solution of the ordinary differential

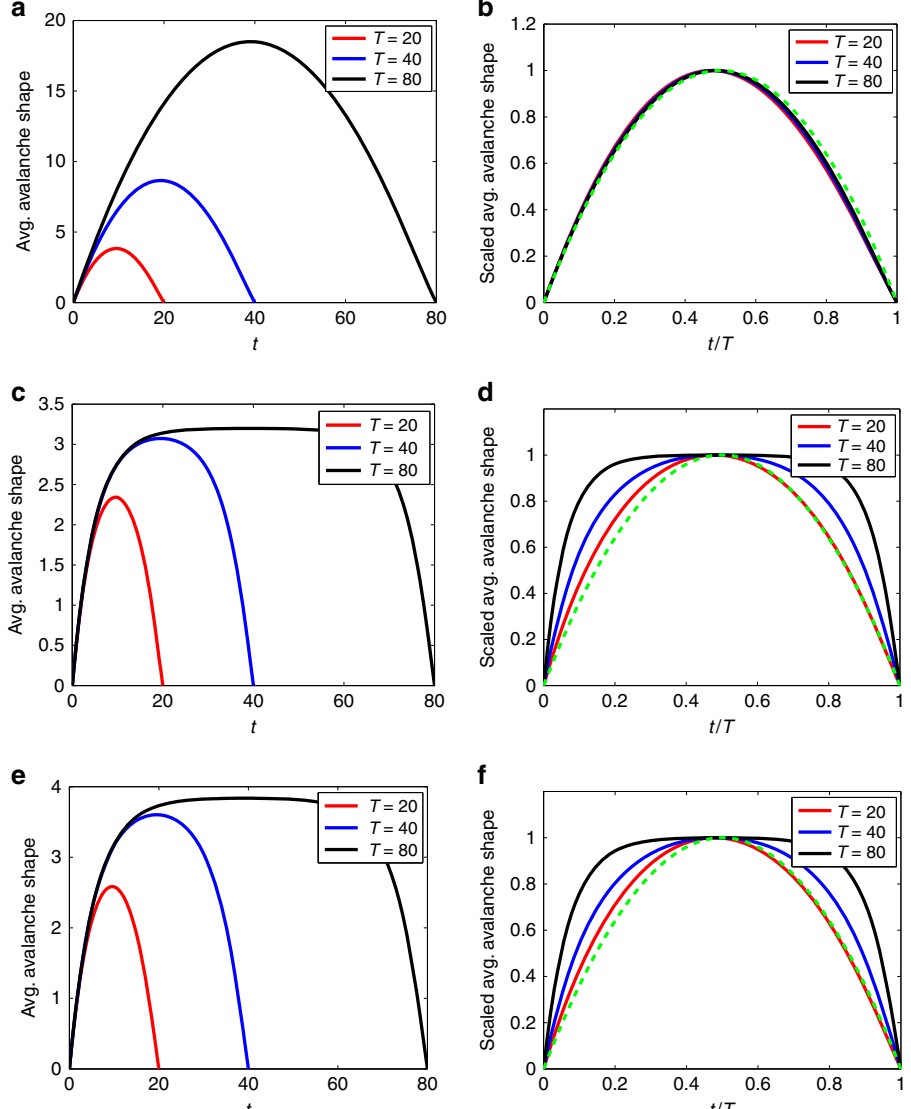

**Fig. 2** Symmetric average avalanche shapes. **a**, **c**, **e** Left column: Average avalanche shapes from Eq. (4), for Poisson offspring distribution $q_k$, with mean $\xi$. Each panel in the right column (**b**, **d**, **e**) shows the same functions as in the panel to its left, but plotted (as, for example, in ref. [7]) in rescaled time $t/T$ and rescaled vertically to have a maximum value of 1. **a**, **b** Critical case with $\xi = 1$; **c**, **d** subcritical case, $\xi = 0.8$; **e**, **f** supercritical case, $\xi = 1.2$. The dashed green curve in **b**, **d**, **f** is the parabola $4\frac{t}{T}\left(1 - \frac{t}{T}\right)$

equation

$$\frac{dQ}{dt} = f(Q) - Q, \quad \text{with } Q(0) = 0. \tag{6}$$

Thus, the average avalanche shape for a given offspring distribution $q_k$ and duration $T$ can be calculated by solving only one ordinary differential equation, Eq. (6), and then using the solution function $Q(t)$ in the explicit formula of Eq. (4). We also show (in Supplementary Note 5) that the average avalanche shape for discrete-time updating can be found in a similar fashion, but the resulting expression is less amenable to analysis than the continuous-time case. However, qualitatively similar results are found for both continuous-time and discrete-time updating (see Supplementary Fig. 2), so we focus mainly on the solution given by Eqs. (4) and (6).

In Fig. 2, we show the avalanche shapes that are given by Eq. (4) in the case where the offspring distribution $q_k$ is a Poisson distribution with mean $\xi$. In the critical case ($\xi = 1$), we see from Fig. 2a, b that a rescaling of time and of avalanche height causes the avalanche shapes for different durations to collapse onto a single symmetric curve: this scaling collapse (although not the shape of the curve) is predicted by the universality arguments of ref. [4]. Note that a case where Eq. (6) is exactly solvable (binary fission) is examined in Supplementary Note 3 and is shown to give symmetric avalanche shapes, which are parabolic at the critical point. For subcritical ($\xi < 1$) avalanches, the profiles are symmetric but non-parabolic (Fig. 2c) and do not collapse onto a universal curve (Fig. 2d). For supercritical ($\xi > 1$) avalanches (where we only consider those avalanches which terminate at a finite time $T$: a non-zero fraction of avalanches also exist that never terminate), very similar symmetric shapes are observed to the subcritical case (Fig. 2e, f).

Next, we consider the case where the offspring distribution has a power-law tail:

$$q_k \sim C k^{-\gamma} \quad \text{as } k \to \infty, \tag{7}$$

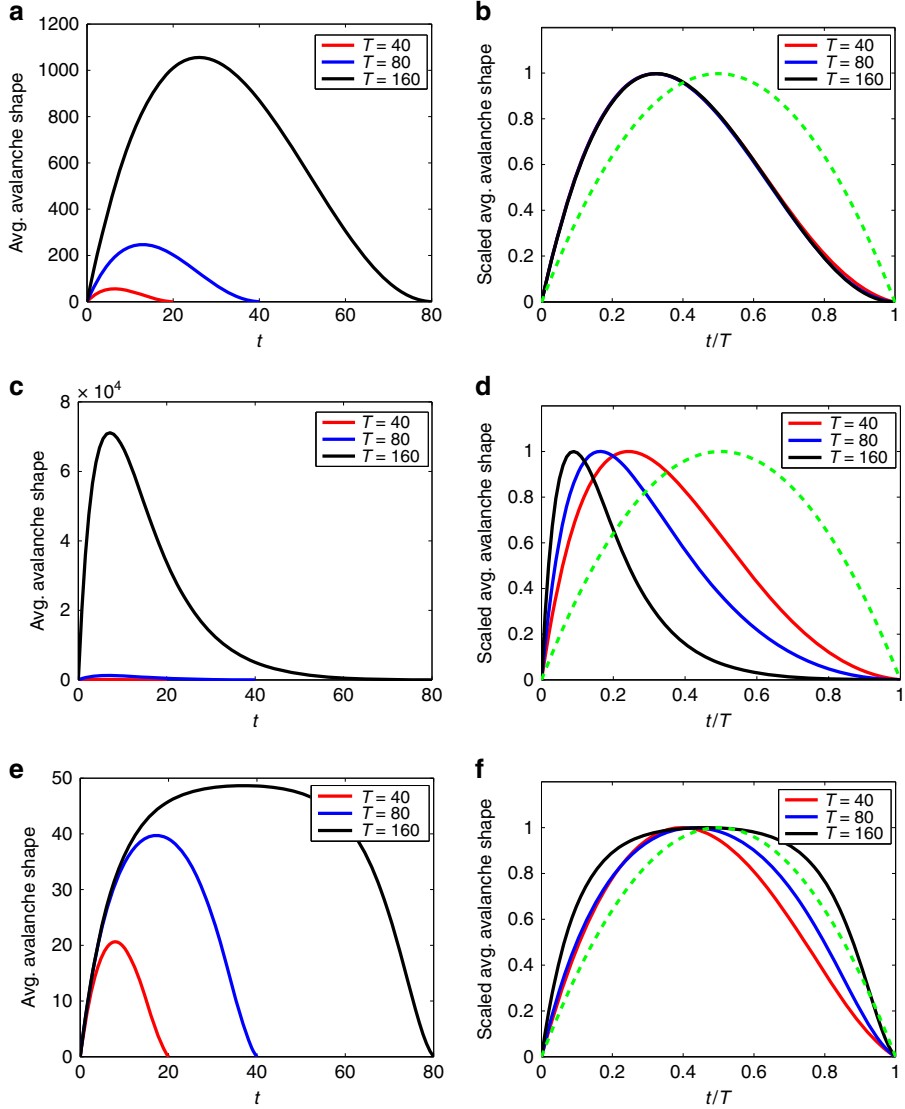

**Fig. 3** Nonsymmetric average avalanche shapes. Left column: Average avalanche shapes from Eq. (4), for power-law offspring distribution: $q_k = C k^{-\gamma}$ for $k \geq 1$, with exponent $\gamma = 2.5$ and constant $C$ chosen to give the branching number $\xi$. Each panel in the right column (**b**, **d**, **e**) shows the same function as in the panel to its left, but rescaled as in Fig. 2. **a**, **b** Critical case with $\xi = 1$; **c**, **d** subcritical case, $\xi = 0.9$; **e**, **f** supercritical case, $\xi = 1.15$; the dashed green curve is the same parabola as in Fig. 2

with exponent $\gamma$ in the range $2 < \gamma < 3$, so that the second moment of the offspring distribution is infinite. (Here, and throughout the paper, we use $C$ to denote a constant prefactor in an asymptotic scaling relation). Such cases are of practical interest because scale-free degree distributions are well known[25, 26, 36] to strongly affect dynamics on networks, and (as we discuss below) a scale-free degree distribution can, for certain dynamics, lead to offspring distributions such as Eq. (7). Note that the second moment of the offspring distribution is related to the second derivative of the generating function of Eq. (5) evaluated at $s = 1$, so this case corresponds to $f''(1) = \infty$.

Using offspring distributions of the form (7) in Eqs. (4) and (6) gives the avalanche shapes shown in Fig. 3. Clearly, the avalanche shapes—both critical and noncritical cases—are nonsymmetric, with a leftward skew. (This contrasts with the right-skewed avalanche shapes found from random-walk models with long memory[14]). A detailed asymptotic analysis of the governing equation (Supplementary Note 6) enables us to conclude that, in the critical case as $T \to \infty$ (and $T - t \to \infty$), the average avalanche

shape scales as:

$$
A(t) \sim
\begin{cases}
C \frac{t}{T}(T - t) & \text{if}\ ''f(1)\ \text{is finite,} \\
C \frac{t}{T}(T - t)^{\frac{1}{\gamma - 2}} & \text{if}\ q_k \propto k^{-\gamma}\ \text{as}\ k \to \infty,\ \text{with}\ 2 < \gamma < 3,
\end{cases}
\tag{8}
$$

where the constant prefactor $C$ is independent of $T$. Note that parabolic avalanche shapes (with peak at $t = T/2$) are seen in the large $T$ limit whenever the offspring distribution $q_k$ has finite second moment, but the shape is nonsymmetric (with peak at $t = (\gamma - 2)T/(\gamma - 1) < T/2$ for power-law distributions with $\gamma$ between 2 and 3.

True power-law tails are never seen in real networks, due to finite-size effects. If the offspring distribution instead has a truncated power-law form, with an exponential cutoff for $k \gg \kappa$:

$$
q_k \sim C k^{-\gamma} e^{-\frac{k}{\kappa}} \quad \text{as}\ k \to \infty,
\tag{9}
$$

and with $2 < \gamma < 3$, then the avalanche shapes for all durations $T$

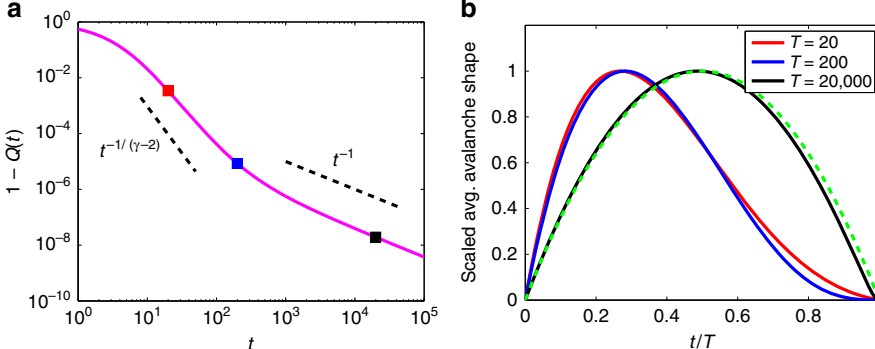

**Fig. 4** Effects of truncated power-law offspring distribution. Survival function $1 - Q(t)$ (**a**) and rescaled avalanche shape (**b**) for offspring distributions with truncated power-law form: $q_k \propto k^{-\gamma} e^{-k/\kappa}$ with $\gamma = 2.3$, $\kappa = 10^6$. The constant of proportionality is determined by the criticality condition $\xi = 1$. The colored squares in **a** mark the durations $T = 20$, $T = 200$, $T = 2 \times 10^4$ of the avalanches whose average shapes are plotted in **b**. Note the $T = 20$ and $T = 200$ cases have the nonsymmetric profiles typical of power-law $q_k$, but the $T = 2 \times 10^4$ case closely matches the parabolic profile expected for offspring distributions with finite second moment; the dashed green curve is the same parabola as in Fig. 2

do not collapse onto a single curve, even at criticality. As the asymptotic analysis of Supplementary Note 6 reveals, the shape for large $T$ is determined by the survival function $1 - Q(t)$: this is the fraction of avalanches that remain alive at a time $t$ after they begin. For the offspring distribution of Eq. (9), the survival function $1 - Q(t)$ scales as $t^{-\frac{1}{\gamma-2}}$ for early times, but as $t^{-1}$ for later times; the crossover from one regime to the other is determined by the exponential cutoff $\kappa$ in the offspring distribution (see Fig. 4a). Therefore, it is possible to observe nonsymmetric shapes for avalanches with relatively short durations $T$, but the longer-duration avalanches (the $T \to \infty$ limit) revert to the parabolic shape typical of offspring distributions with finite second moment, see Fig. 4b.

**Other characteristic temporal shapes**. The Markov branching process approach that we use to derive the average avalanche shape in Eq. (4) can also be applied to calculate other temporal characteristics of avalanches. In Supplementary Note 2, for example, we derive a formula for the variance of the avalanche shape (i.e., the variance of the set of functions $\{V_a(t)|a \in S_T\}$, see "Methods" section). The overall shape of the standard deviation is found to be similar to the average avalanche shape: The asymptotic result for the critical case (in the limit $T \to \infty$ and $T - t \to \infty$, see Supplementary Note 6) is that the coefficient of variation at time $t$ for avalanches of duration $T$ is

$$\mathrm{CV}(t) = \frac{\sqrt{\text{variance}}}{A(t)} \sim \begin{cases} \frac{1}{\sqrt{2}} & \text{if } f''(1) \text{ is finite,} \\ \sqrt{\frac{3-\gamma+(\gamma-2)\frac{t}{T}}{(\gamma-1)\frac{t}{T}}} & \text{if } q_k \propto k^{-\gamma} \text{ as } k \to \infty, \text{ with } 2 < \gamma < 3. \end{cases}$$

(10)

Note that the coefficient of variation is constant (independent of $t$ and $T$) for the case where parabolic avalanche shapes occur, so the standard deviation also has a parabolic profile. However, in the power-law case with $\gamma < 3$, the shape of the standard deviation is even more skewed than that of the average avalanche shape profile: the coefficient of variation limits to a constant as $t \to T$, but it diverges like $1/\sqrt{t}$ as $t \to 0$.

The universality of the average avalanche shape has been used in analyzing experimental data; specifically, the collapse of shapes for avalanches of different durations can help identify whether the dynamics is critical or not[3, 4, 7, 9]. However, one drawback of the average avalanche shape is that it requires an accurate assessment of the time $T$ at which each avalanche terminates. Pinpointing such termination times can be difficult in empirical data, especially for avalanches of information on social networks,

many of which exhibit very long lifespans[34, 37, 38]. Another characteristic temporal shape that we can calculate analytically is the average shape of all avalanches that have not terminated by a given observation time $T$: such a characteristic may prove easier to calculate for empirical data than the standard avalanche shape. In Supplementary Note 4, we show that the average non-terminating avalanche shapes at various observation times $T$ collapse onto a single curve when the dynamics are critical. The universal curve is again found to have a parabolic form (but with peak at $t = T$) if $f'(1)$ is finite, and to have a skewed (non-parabolic) shape if the offspring distribution is power-law. Moreover, the calculation of the average non-terminating avalanche shape requires less data than that of the average avalanche shape (see Supplementary Note 7), which may make it a useful diagnostic tool in experimental studies.

As we demonstrate with our numerical simulations below, an even simpler temporal profile can provide a very sensitive measure of whether an avalanching system is critical or not. The average number of events observed at a time $t$ after an avalanche begins is given by the average of $V_a(t)$ over the entire set of avalanches, regardless of the duration of the avalanche. (Note avalanches that terminated at a time $T$ with $T < t$ contribute zero to the measure at time $t$.) In Supplementary Note 4, we show that the average number of such events at time $t$ is an exponentially decaying function of $t$ if the dynamics is subcritical, an exponentially growing function of $t$ for supercritical dynamics, and is a constant (independent of $t$) for the critical case. This temporal characteristic is particularly useful in understanding the numerical simulation results below, and it can equally well be applied to experimental data.

**Numerical simulations**. In this section we report on numerical simulations of unidirectional dynamics on networks, to assess the applicability of the branching process theory developed above. We run numerical simulations of example dynamics on synthetic (configuration-model) and real-world networks, recording average avalanche shapes (and other temporal characteristics) for comparison with our theory. The branching process paradigm is only an approximation for dynamics on networks, as its assumptions are invalidated by the existence of loops in the network and by the finite size of the networks[26]. We demonstrate that while such features indeed impact upon the agreement with theoretical results, the main qualitative feature of critical dynamics that we have identified—the appearance of nonsymmetric avalanches shapes when the offspring distributions defined

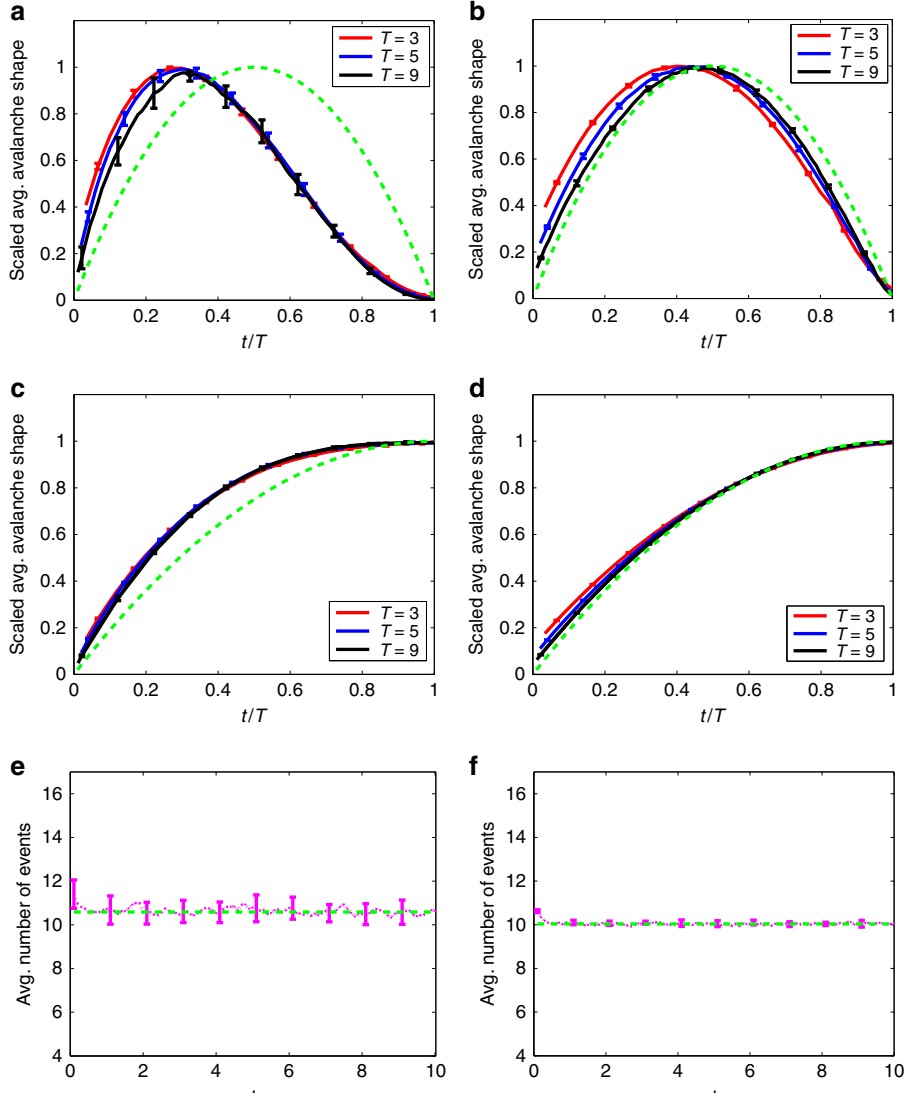

**Fig. 5** Twitter cascades model. Numerical simulation results for critical meme-popularity avalanches within the model of ref. [32]. The panels in the left column show results from a network with a power-law out-degree distribution, with exponent $\alpha = 2.5$. The results in the right column are from a network where every node has exactly $z = 10$ out-neighbors. **a, b** Rescaled average avalanche shapes. **c, d** Rescaled average non-terminating avalanche shapes. **e, f** Average number of tweets of the meme per unit time. Here and in Figs. 6–8, the green dashed curve in **a, b** is the parabola described in Fig. 2, while that in **c, d** is the half-parabola $\frac{t}{T}\left(2 - \frac{t}{T}\right)$. In **e, f**, the green dashed line is the constant value representing the late-time average value of the number of events (the average is taken over the second half of the time range, i.e., $t$ values between 5 and 10). See "Methods" section for definition of error bars

by Eq. (1) or (2) have sufficiently heavy tails—is indeed observable in numerical simulations of cascade dynamics on networks.

Figure 5 shows results from simulations of the continuous-time (Markovian) meme propagation model of ref. [32]. Memes are tweeted from user to user according to the rules of the model; the popularity of each meme is tracked in the simulations, and the number of events in the avalanche profile of a meme is the number of times it is tweeted within a time interval. We record the average avalanche shape determined by all memes whose avalanches terminate at time $T$ (using a bin of duration 0.5, so $T = 9$, e.g., includes all avalanches that terminate at times in the range (8.5, 9)). Two network structures are compared (see "Methods" section): the panels in the left column of Fig. 5a, c, e present results for a network with scale-free out-degree distribution $p_k \propto k^{-\alpha}$ with $\alpha = 2.5$, while the panels in the right column (Fig. 5b, d, f) are for a network where every node has exactly $z = 10$ followers (note the mean degree of the two networks are approximately equal). In both cases, the in-degree distribution is

Poisson, and in-degrees and out-degrees of nodes are independent. We set the innovation parameter $\mu$ to zero, so we expect the dynamics to be critical (from Eqs. (2), (3), and (13)).

According to our theory, the rescaled average avalanche shape curves for different (and sufficiently large) avalanche durations $T$ should collapse onto a single curve. We see good agreement with this prediction in Fig. 5a: note the distinctively nonsymmetric shape of the collapsed curve. Although the average avalanche shapes for the $z$-regular out-degree network in Fig. 5b are not fully converged by $T = 9$, they are evidently approaching the parabolic profile expected for the case where the offspring distribution has finite second moment. Figure 5c, d contrast the average non-terminating avalanche shapes that are found on the two networks. On the power-law network, the shape is clearly non-parabolic (Fig. 5c), while the match to the asymptotic expression found in Supplementary Note 6 is very good for the case of finite $f''(1)$ (Fig. 5d). On both types of network, the error bars (see "Methods" section for definition) are smaller than in

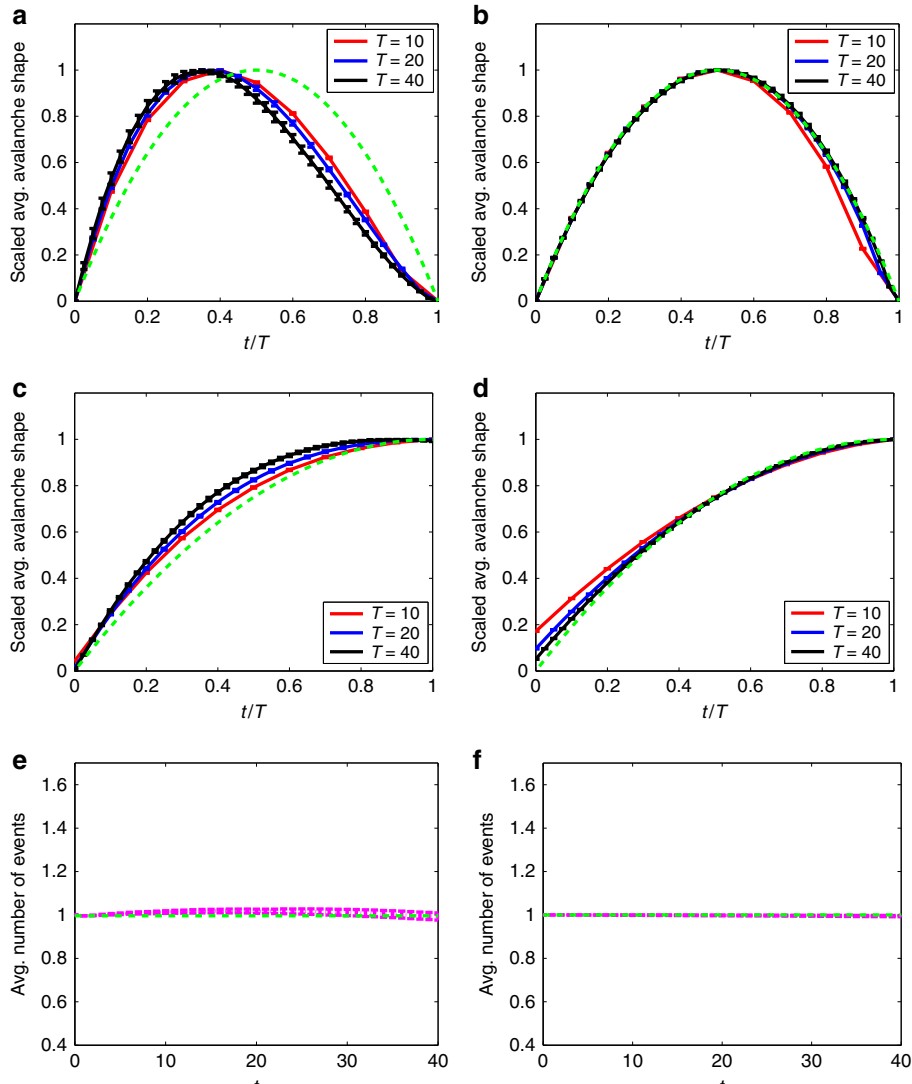

**Fig. 6** Neuronal avalanches model. Results of numerical simulation of the neuronal dynamics model of ref. [3] at criticality. As in Fig. 5, the results in the left column are for a network with power-law out-degree distribution ($\alpha = 2.5$), while those in the right column are for a network with $z$-regular out-degrees. **a**, **b** Rescaled average avalanche shapes. **c**, **d** Rescaled average non-terminating avalanche shapes. **e**, **f** Average number of firing neurons per discrete-time step. The green dashed line in **e**, **f** shows the constant value 1 that is expected for the average number of events in this critical discrete-time branching process. See "Methods" section for definition of error bars

Fig. 5a, b, reflecting the fact that the set of avalanches that have not terminated by time $T$ is typically much larger than the set of avalanches that terminate exactly at $T$, so the average shape is better estimated using the larger data set. Figure 5e, f shows the average number of events (tweets) per unit time. The criticality of the process on both networks is reflected in the fact that this measure remains constant over time.

Figure 6 shows results obtained from simulations of the neuronal dynamics model of ref. [3], where the parameter $\phi_{\max}$ is tuned so as to poise the dynamics near to criticality. As in Fig. 5, the left column of results is for a network with power-law out-degree distribution of exponent $\alpha = 2.5$, while those in the right column are for a $z$-regular out-degree network. All neurons are synchronously updated in each discrete-time step and we record the number of activated neurons at each step as the "events" of the avalanche; the avalanche terminates when no new neurons are activated. Although the details of this discrete-time case differ from the continuous-time case of Fig. 5, the results again qualitatively agree with theory. As expected (see "Methods" section), we see nonsymmetric average avalanche

shapes and non-parabolic average non-terminating avalanche shapes in Fig. 5a, c, while the corresponding results on the finite second moment network (Fig. 5b, d) agree closely with the parabolic asymptotic shapes of Eq. (8) and Supplementary Note 6.

In Fig. 7, we consider threshold dynamics on an undirected network. Specifically, we show results for a Centola–Macy threshold model, with a uniform distribution of thresholds on the interval $(0, \theta_{\max})$, where the parameter $\theta_{\max}$ is tuned to place the dynamics close to criticality. The updating is synchronous, i.e., all nodes are updated at each discrete-time step, and the number of avalanche events at each time step is the number of nodes that are newly activated. For the Centola–Macy dynamics, we expect (see "Methods" section) nonsymmetric average avalanche profiles when the power-law exponent $\alpha$ of the network degree distribution lies between 3 and 4; note we use $\alpha = 3.3$ in the left column of Fig. 7. It is noteworthy that the network degree distribution has finite variance in this case: it is the interaction between the network topology and the Centola–Macy dynamics that leads to nonsymmetric avalanche

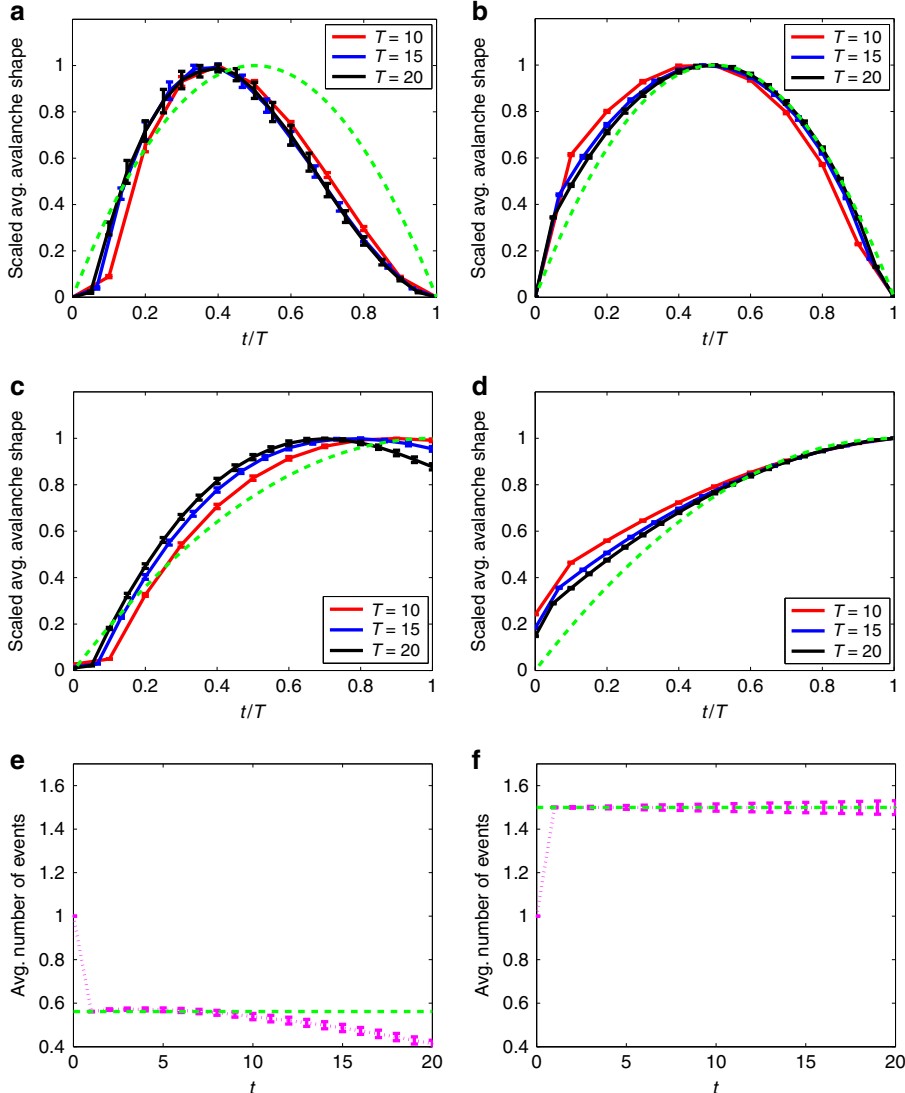

**Fig. 7** Behavior adoption model. Results of numerical simulation of a Centola–Macy threshold model[29] at criticality. Left column results are for a scale-free network ($\alpha = 3.3$); right column results are for a $z$-regular random network. **a**, **b** Rescaled average avalanche shapes. **c**, **d** Rescaled average non-terminating avalanche shapes. **e**, **f** Average number of newly activated nodes per discrete-time step. The green dashed line in **e**, **f** shows the constant value $z^2/\langle k^2 - k \rangle$ expected for the average number of events at criticality (see "Methods" section). See "Methods" section for definition of error bars

shapes. (In contrast, for the Watts threshold model with uniformly distributed thresholds, nonsymmetric profiles appear only for networks with degree exponents $\alpha$ in the range $2 < \alpha < 3$, i.e., for degree distributions with infinite variance).

In this case, we can clearly see one of the limitations of the branching process approximation: the seed node for each cascade is chosen uniformly at random from all the nodes, but subsequently activated nodes in the cascade are reached with a probability proportional to their degree, as in Eq. (1). Therefore, the branching process picture does not correctly capture the first step of the cascade and this discrepancy can be seen in the early time shape of the avalanches, particularly in Fig. 7a, b. The theory could be extended to deal with this issue (as in ref. [39] for example), but the effects on the average profiles diminish as longer-duration avalanches are considered.

A more serious limitation of the theory's accuracy is presented by Fig. 7e, where we see the deviation of the average number of events away from the constant value that indicates critical dynamics. In fact, the finite size of the network and the heavy-

tailed degree distribution mean that the activated nodes are quickly (within about 10 time steps) replacing inactive nodes throughout a significant fraction of the network, so that some of the "new branches" emanating from an activated node are in fact connected to previously activated nodes, contrary to the branching process assumption of independence. As a result, the spreading efficiency decreases over time, and the branching process—despite initially being at criticality—becomes subcritical. Nevertheless, as Fig. 7a shows, the average avalanche shapes still collapse quite well, and clearly are nonsymmetric; the average shape of non-terminating avalanches (in Fig. 7c) is a more sensitive indicator of the loss of criticality due to the finite size of the network.

All networks used in the simulations above were configuration-model networks. However, real-world networks are known to differ significantly from configuration-model networks with the same degree distribution[26]. For example, degree–degree correlations and closed triangles of nodes ("clustering") are common in many social networks, but are relatively rare in the corresponding

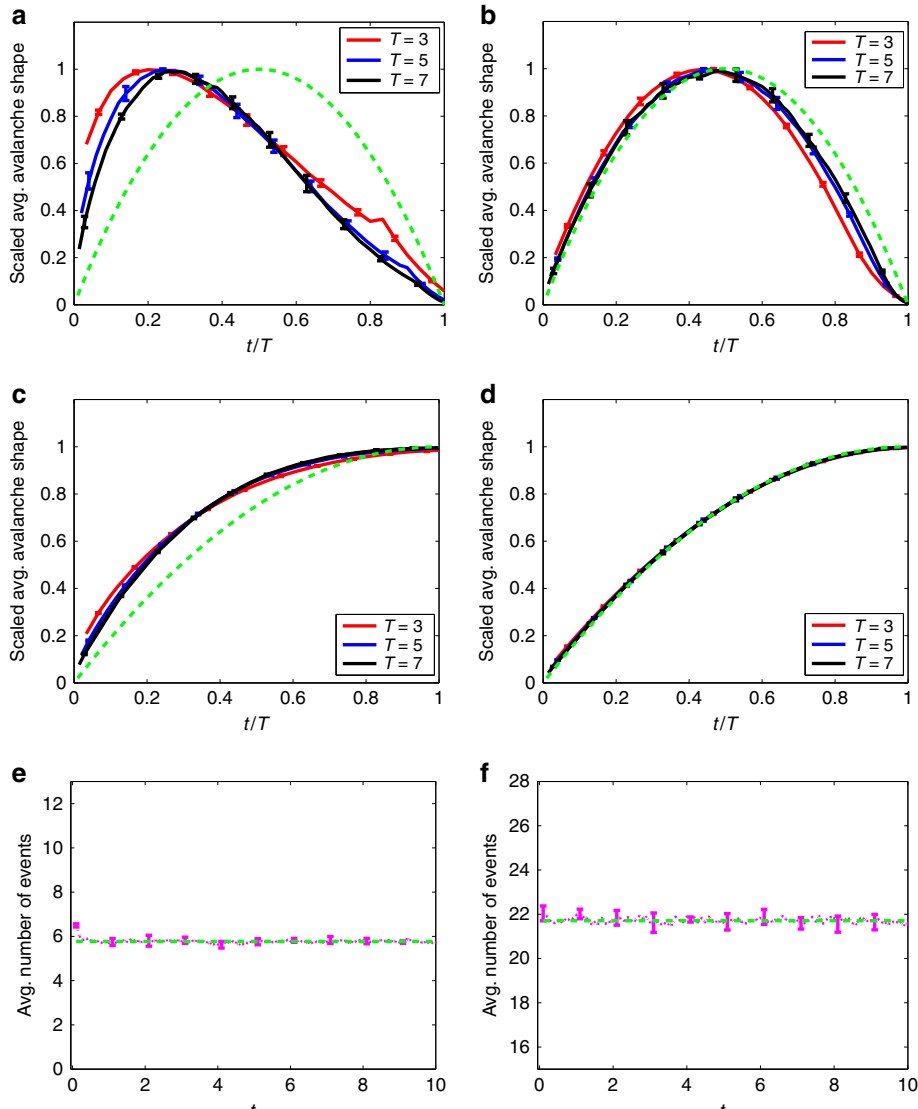

**Fig. 8** Cascades on empirical Twitter network. Numerical simulation results for critical meme-popularity avalanches[32]. The results in the left column are for the Twitter network of ref. [40], while those in the right column are for a directed Erdös–Rényi network with the same mean degree as the Twitter network (see Supplementary Note 8). Although the empirical network is not tree-like, the qualitative predictions of our theory still hold: compare to the corresponding panels of Fig. 5. Panels (**a**) and (**b**): rescaled average avalanche shapes. Panels (**c**) and (**d**): rescaled average non-terminating avalanche shapes. Panels (**e**) and (**f**): average number of tweets of the meme per unit time. The green dashed curves are defined as in Fig. 5. See "Methods" section for definition of error bars

configuration-model networks. In our final example, we therefore use the sample of the Twitter network that is made available for download by the authors of ref. [40]. Unlike a configuration-model-directed network, this data set contains a high proportion of reciprocated links (i.e., instances where node $i$ follows node $j$ and node $j$ also follows node $i$); in fact, 48% of all links in the network are reciprocated. This feature means that the network is not tree-like, and the branching process assumptions are not strictly true. Nevertheless, Fig. 8 shows that results from numerical simulations of the critical meme propagation model of ref. [32] are in very good qualitative agreement with the predictions of our theory: a good collapse of the average avalanche shapes for different durations $T$ is found (Fig. 8a), with a clear left skew that is consistent with the heavy-tailed distribution of number of Twitter followers found in empirical data[34, 41]. By randomly rewiring the original network[42], we confirm that the left skew

is due to the degree distribution of the network, and not to any meso- or macro-scopic structure of the network, see Supplementary Note 8. As in other examples, where tree-based theory is more accurate than expected on real-world networks[43], this result demonstrates that the predictions of the theory are quite robust to violations of the assumptions used in the mathematical derivation.

Another important assumption of the mathematical derivation is the Markovian nature of the dynamics. In Supplementary Note 9, we generalize the meme propagation model to include non-Markovian dynamics[39, 44], so that the inter-event times between successive tweets of a user need not be exponentially distributed. Although we are limited to simulation results in this case, the qualitative predictions of the Markovian theory presented here again appear to be robust even to quite strongly non-Markovian dynamics.

## Discussion

In this paper we have examined the link between Markov branching processes and unidirectional cascade dynamics on networks, with a focus on the temporal profile of avalanches. Our main result is Eq. (4), which gives the average avalanche shape for avalanches (both critical and noncritical) of duration $T$, and requires only the solution of a single ordinary differential equation (Eq. 6). The input to this equation is the offspring distribution of the branching process, which is determined from the network structure and the dynamical system of interest by Eq. (1) or (2). In our analysis of the avalanche shape given by Eq. (4), we have demonstrated that nonsymmetric avalanche shapes can arise at criticality when the offspring distribution has a power-law tail. Using numerical simulations of threshold models, neuronal dynamics, and online information-sharing, we show that nonsymmetric shapes can occur for common models running on networks with power-law degree distributions. However, it is important to note that a heavy-tailed degree distribution is not sufficient to guarantee nonsymmetric avalanche shapes: It is the interplay between the cascade dynamics and the network topology (as can be seen from the formulas in Eqs. (17) and (20) for the exponent of the offspring distribution) that determines the symmetry of the average avalanche shape. The results of numerical simulations verify the qualitative predictions of the branching process theory, despite finite-size effects and other violations of the assumptions of the theory. Further simulation studies (Supplementary Note 9) indicate that the qualitative results remain valid even for non-Markovian dynamics, although further theoretical investigation is clearly required to verify the regimes of validity.

In addition, our theoretical approach enables us to identify other characteristic temporal shape functions (e.g., the average non-terminating avalanche shape, see Supplementary Note 4) that may prove useful when experimentalists seek to identify critical behavior from the temporal signatures in a data set with a limited number of avalanche time series. Given the relevance of branching process descriptions to cascades in a range of fields (e.g., neuroscience[45], social networks[34], crackling noise[4], etc.), it is hoped that these insights may find many applications. We anticipate several possible directions for extensions of the methodology introduced here; notably, removing the Markovian assumption to apply a similar analysis for non-Markovian cascades[34],[39], and extending the theory to multilayer networks[46],[47].

## Methods

**Vulnerabilities**. Here we give examples of how the vulnerabilities $v_k$ and $v_{jk}$ are calculated for the models introduced in the main text. Inserting these $v_k$ and $v_{jk}$ functions into Eqs. (1) and (2), respectively, defines the offspring distribution for each of the models.

The vulnerability $v_k$ for a threshold model on an undirected network is the probability that a node $i$, of degree $k$, activates when exactly one of its neighbors is active, i.e., when $m_i = 1$. According to the rules of the Watts threshold model, for example, node $i$ will become active if its threshold $R_i$ is less than, or equal to, $1/k$, which is the fraction of its neighbors that are active when $m_i = 1$. The probability that the node's threshold is less than this value is given by:

$$v_k^{(\text{Watts})} = F\left(\frac{1}{k}\right), \tag{11}$$

where $F$ is the cumulative distribution function of the thresholds. In the Centola–Macy threshold mode, a node with one active neighbor will activate if its threshold is less than or equal to 1, and so the vulnerability in this case is

$$v_k^{(\text{C--M})} = F(1). \tag{12}$$

In the neuronal dynamics model of ref. [3], a node is vulnerable if it becomes active when one of its in-neighbors fires. According to the rules of the model, this occurs with a probability that depends on the edge between the two nodes, but is chosen from a uniform distribution on $(0, \phi_{\max})$, independently of the degrees of

the nodes. The probability that a random node with in- and out-degrees $j$ and $k$ is vulnerable is therefore the average of this uniform distribution, i.e., $v_{jk} = \phi_{\max}/2$.

In the model of ref. [32] for meme propagation on a directed social network, we focus on a chosen meme and assume that this meme has been tweeted by an in-neighbor of node $i$ (i.e., by one of the $j$ users followed by user $i$). The probability that node $i$ will subsequently retweet this meme (before user $i$'s memory is overwritten by other tweets it receives from the $j$ users it follows) is given, for $j \gg 1$, by:

$$v_{jk} \approx \frac{1 - \mu}{j}, \tag{13}$$

where $\mu$ is the innovation probability and $k$ is the number of followers of node $i$, see Sec. IVA of ref. [34].

**Power-law offspring distributions from network dynamics**. In Supplementary Note 6, we show that nonsymmetric average avalanche shapes occur within Markovian branching processes when the offspring distribution $q_k$ has a power-law tail with exponent $\gamma$ between 2 and 3. Here we examine how such offspring distributions might arise from unidirectional dynamics on undirected and directed networks, using the relationships given by Eqs. (1) and (2).

In the case of an undirected network, we suppose that the vulnerability $v_k$ depends on the node degree $k$ as:

$$v_k \sim C k^{-\nu} \text{ as } k \to \infty. \tag{14}$$

Then, if the degree distribution of the network has a power-law tail:

$$p_k \sim C k^{-\alpha} \text{ as } k \to \infty, \tag{15}$$

the large-$k$ asymptotics of $\hat{q}_k$ (and hence of the offspring distribution $q_k$, see Supplementary Note 1) are given by Eq. (1) as:

$$q_k \sim C k^{1-\alpha-\nu} \tag{16}$$

and so we write

$$\gamma^{(\text{undir})} = -1 + \alpha + \nu \tag{17}$$

for the power-law exponent of the offspring distribution. Note that in general, the value of $\gamma$ will be different from the power-law exponent $\alpha$ of the network's degree distribution.

The case of a directed network is complicated by the existence of the joint distribution $p_{jk}$ of in-degree $j$ and out-degree $k$. If we assume the simplest case of nodes having independent in- and out-degree, then the joint distribution factorizes: $p_{jk} = p_j^{\text{in}} p_k$, and the large-$k$ behavior of the vulnerability can be specified by the weighted sum over in-degrees as:

$$\sum_j \frac{j}{z} p_j^{\text{in}} v_{jk} \sim C k^{-\nu}. \tag{18}$$

Assuming a power-law out-degree distribution, as in Eqs. (2) and (15) and the equivalence of the power-law exponents of $q_k$ and $\hat{q}_k$ yields

$$q_k \sim C k^{-\alpha-\nu} \text{ as } k \to \infty, \tag{19}$$

so that the power-law exponent of the offspring distribution is

$$\gamma^{(\text{dir})} = \alpha + \nu. \tag{20}$$

**Details of numerical simulations**. Each of Figs. 5–7 consists of two columns of panels. The left-hand column (i.e., Figs. 5a, c, e and 7a, c, e) show results for a network with a power-law degree (or out-degree) distribution $p_k \propto k^{-\alpha}$ for $k \geq k_{\min}$ (with $p_k = 0$ for $k < k_{\min}$). The right-hand column (Figs. 5b, d, f and 7b, d, f) are for random regular networks, where every node has the same (out-)degree $z$. In each experiment, $n_A$ individual avalanches are simulated to calculate the average avalanche shape and other measures (see Supplementary Note 7 for a study of how the value of $n_A$ impacts upon the results shown). For each avalanche, the seed node is changed and the order of node updates (for Figs. 5 and 8) or the dynamical parameters (the edge weights $\phi_{ij}$ for Fig. 6 or the node thresholds $R_i$ for Fig. 7) are randomized. In order to quantify the robustness of the results, the complete experiment that results in the average avalanche shape is repeated a total of $n_R$ times and the error bars in the figures denote the standard deviation of the measures over the set of replica experiments.

Figure 5 shows results for the meme propagation model of ref. [32] at criticality (the innovation parameter $\mu$ is zero), using $n_A = 10^6$ avalanches and $n_R = 12$ replicas, on networks with $N = 10^5$ nodes. The directed network with power-law out-degree distribution (Fig. 5a, c, e) has exponent $\alpha = 2.5$ and minimum out-degree of $k_{\min} = 4$, giving a mean degree of $z = 10.6$. The panels in the right column are for a directed network where every node has exactly $k = 10$ followers. In each case, the followers are chosen uniformly at random from the set of all nodes, so in-

degrees and out-degrees are independent. Using the vulnerability from Eq. (13) in Eq. (18) gives a $\nu$ value of 0 for this model, so Eq. (20) predicts nonsymmetric avalanche shapes for values of $\alpha$, the tail exponent of the network's out-degree distribution, between 2 and 3; note we use $\alpha = 2.5$ in the left column of Fig. 5.

The neuronal dynamics model of ref. [3] is used for the simulations of Fig. 6, with the parameter $\phi_{max}$ set to its critical value of $2/z$. Here the vulnerability $v_{jk} = \phi_{max}/2$ is independent of node degree $k$, and so $\nu = 0$ in Eq. (18), giving $\gamma = \alpha = 2.5$ from Eq. (20) for the power-law network. We use $n_A = 10^7$ avalanches and $n_R = 24$ replicas, on the same directed networks as used in Fig. 5.

Figure 7 is for the Centola–Macy threshold model with thresholds uniformly distributed between 0 and $\theta_{max} = \langle k^2 - k \rangle/z$. Using $n_A = 10^6$ avalanches and $n_R = 24$ replicas, we run simulations on scale-free undirected networks with $\alpha = 3.3$, $k_{min} = 2$, $z = 2.9$, and $N = 10^6$ (left column panels) and on random $z$-regular graphs with $z = 3$ and $N = 10^5$ (right column panels). The vulnerability $v_k$ of Eq. (12) is a constant in this case, so $\nu = 0$ in Eq. (14). As a result, Eq. (17) gives $\gamma = -1 + \alpha$ as the exponent of the offspring distribution, and so nonsymmetric avalanche shapes are expected for power-law networks with exponent $\alpha$ between 3 and 4 (as in the left column of Fig. 7). However, note that if we instead consider the Watts threshold model with uniformly distributed thresholds, Eq. (11) gives $\nu = 1$, hence $\gamma = \alpha$, meaning that nonsymmetric avalanche shapes occur only for networks with degree exponents $\alpha$ in the range $2 < \alpha < 3$. The differing conditions for the two threshold models provide a good example of how the node-to-node dynamics and the network topology interact in a nontrivial fashion to generate nonsymmetric average avalanche shapes.

The green dashed lines of Fig. 7e, f show the expected number of "children" events triggered by a seed node that is chosen uniformly at random (not with probability proportional to its degree, as in Eq. (1)). The seed node has an average of $z$ neighbors, each of which activates with probability $F(1) = 1/\theta_{max}$, giving an expected number of events (after the seeding event at $t = 0$) equal to $z^2/\langle k^2 - k \rangle$.

In Fig. 8, we use the same meme propagation model as in Fig. 5 (with $\mu = 0$), running $n_A = 1.4 \times 10^6$ avalanches in $n_R = 6$ replicas. The network substrate is the sampled Twitter network of ref.[40], which has mean degree $z = 21.75$ and $N = 81,306$ nodes.

**Code availability**. Matlab/Octave simulation codes for the examples used in this paper are available from http://www.ul.ie/gleesonj/avalanches.

**Data availability**. Network data for the examples used in this paper are available from http://www.ul.ie/gleesonj/avalanches. The empirical Twitter network used in Fig. 8 is available from the SNAP repository http://snap.stanford.edu/data/egonets-Twitter.html.

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

## Acknowledgements

Helpful discussions with Gareth Baxter, Sergey Dorogovtsev, Ali Faqeeh, Peter Fennell, Tom Hurd, Kristina Lerman, Kevin O'Sullivan, Mason Porter, and Tim Rogers are

gratefully acknowledged. This work was supported by the Science Foundation Ireland (Grant numbers 11/PI/1026, 09/SRC/E1780, 15/SPP/E3125, and 16/IA/4470). We acknowledge the SFI/HEA Irish Centre for High-End Computing (ICHEC) for the provision of computational facilities.

## Author contributions

J.P.G. and R.D. conceived the project and wrote the manuscript. J.P.G. developed analytical results and performed numerical simulations.

## Additional information

**Competing interests:** The authors declare no competing financial interests.

