## [Peer Review File · Nature Communications]

Reviewers' comments:

Reviewer #1 (Remarks to the Author):

The authors study theoretically the average shape of fixed-duration avalanches in networks that satisfy the locally tree-like assumption. They find that the shape is universal at criticality, and that this shape can become asymmetric depending on the interplay between the network structure and the avalanche dynamics. The authors also present asymptotic results for large duration avalanches and a derivation of the average number of excitations for non-terminating avalanches. The predictions from the theory are compared with numerical simulations on synthetic and real-world networks. The results shed light on previous experimental and numerical observations, and are a timely addition to the set of theoretical tools for the study of avalanche propagation in neural and social networks. The authors also propose straightforward experimental tests to quantify whether a system is in the critical state. The paper is clearly written and I strongly recommend publication. My only comment is that the caption of Figs. 4 - 8 should describe what the green dashed line is.

Reviewer #2 (Remarks to the Author):

In this manuscript the authors study and interesting but highly stylized feature of cascade propagation in social networks: the fact that, at the critical point, the average temporal profile of the cascades can be rescaled to a universal shape for t/T for different T (T being the final time of the cascade). To do that, the authors solve analytically the case of cascade dynamics in networks using the local tree approximation of configuration-model networks. It is found that scale-free networks have a different scaling function (skewed universal shape) compared to homogeneous networks (parabolic shape)

Although I found the result technically sound, I have major concerns about the importance, impact and potential applicability of the results for the potential reader of Nature Communications.

Specifically:

1. The results only apply for the critical point of the branching process, i.e. it only works for a very special case of the parameters in the model and network. Most cascades in social networks happen away from that critical point. In that case, the avalanche profile is not universal in the t/T variable
2. Results are only for networks without clustering, degree correlations and/or communities. However, it is known that mesoscopic structure like clustering, communities can slow down propagation of cascades (see Karsai, M., Kivela, M., Pan, R. K., Kaski, K., Kertesz, J., Barabasi, A.-L., & Saramaki, J. (2011). Small But Slow World: How Network Topology and Burstiness Slow Down Spreading, 83(2), 025102.) So it might be that long-ranged temporal profile of the cascades is more impacted by the meso and macro structure of the network.
3. Results are obtained assuming that the temporal dynamics of propagation is Poissonian. However, it is well known that inter-event distributions are not Poissonian and that this yields to a different dynamics. Specifically, non-Poissonian inter-event distributions modify the temporal profile of average branching process so most likely are going to modify that of the cascades (see Iribarren, J. E. L., & Moro, E. (2009). Impact of human activity patterns on the dynamics of information diffusion, 103(3), 038702–038702. <http://doi.org/10.1103/PhysRevLett.103.038702> and [38])
4. Results are presented only for the average temporal profile. But how many cascades do we have to observe to reproduce the average temporal profile? As it is shown in figure 1, actual single cascade profiles can be very different from the average temporal profile. By the way, the article does not mention how many cascades are used to obtain the average temporal profiles. This is an

important point, since it seems that from equation (10), the coefficient of variation diverges as $1/\sqrt{t}$ for small t (in the case of scale-free networks).

5. Results are presented mainly for synthetic networks and only a final comparison of the meme propagation model on one instance of empirical Twitter network is done. Although the authors claim their results justify the predictions of the theory, one might wonder what is the statistical significance of this claim. That is, if we are only given the information displayed in figure 8b, can one establish that the model is mainly explaining what we are observing?

6. In this regard, the paper lacks an important point for the community: what is the applicability of the results presented? Do the authors already know of or have in mind any domain in which those results can have an important impact? The spreading of information cascades is a highly active field (the authors have work on that) and many datasets have been collected of hashtags spreading in Twitter, links in blogs, news in Facebook, etc. Some of those datasets are available and the authors could have applied their theory to see if it applies to any experimental result.

Given the points above, I found the results presented very interesting, but perhaps only for a specialized community working on theoretical aspects of cascades dynamics.

Reviewer #1 (Remarks to the Author):

The authors study theoretically the average shape of fixed-duration avalanches in networks that satisfy the locally tree-like assumption. They find that the shape is universal at criticality, and that this shape can become asymmetric depending on the interplay between the network structure and the avalanche dynamics. The authors also present asymptotic results for large duration avalanches and a derivation of the average number of excitations for non-terminating avalanches. The predictions from the theory are compared with numerical simulations on synthetic and real-world networks. The results shed light on previous experimental and numerical observations, and are a timely addition to the set of theoretical tools for the study of avalanche propagation in neural and social networks. The authors also propose straightforward experimental tests to quantify whether a system is in the critical state. The paper is clearly written and I strongly recommend publication. My only comment is that the caption of Figs. 4 - 8 should describe what the green dashed line is.

Reply to Reviewer #1: We thank Reviewer #1 for his/her positive comments and recommendation for publication, and also for highlighting the importance of the experimental tests that we propose (we discuss this in more detail in Reply 6 to Reviewer #2 below). As suggested, we have added text to the captions of Figs. 4 through 8 to describe the green dashed curves and lines.

Reviewer #2 (Remarks to the Author):

In this manuscript the authors study and interesting but highly stylized feature of cascade propagation in social networks: the fact that, at the critical point, the average temporal profile of the cascades can be rescaled to a universal shape for t/T for different T (T being the final time of the cascade). To do that, the authors solve analytically the case of cascade dynamics in networks using the local tree approximation of configuration-model networks. It is found that scale-free networks have a different scaling function (skewed universal shape) compared to homogeneous networks (parabolic shape)

Although I found the result technically sound, I have major concerns about the importance, impact and potential applicability of the results for the potential reader of Nature Communications. Specifically:

1. The results only apply for the critical point of the branching process, i.e. it only works for a very special case of the parameters in the model and network. Most cascades in social networks happen away from that critical point. In that case, the avalanche profile is not universal in the t/T variable

Reply to point 1: While it is true that most of our focus in this paper is on avalanche shapes at criticality, we point out that the main result of the paper is equation (4), which gives the average avalanche shape for both critical and noncritical dynamics, and so is not restricted to analysis near the critical point. In Figure 2 (panels (c) through (f)) we illustrate examples of using this equation for

subcritical and supercritical cases for a Poisson offspring distribution, and in the revised manuscript we have extended Figure 3 so that the subcritical and supercritical cases for a power-law offspring distribution are similarly illustrated. We have also extended Supplementary Note 3 to give the exact solution for binary fission in noncritical as well as critical cases: we find (as in Fig. 2) that the average avalanche shape functions are symmetric about $t=T/2$ in all cases, and precisely parabolic at criticality. Please also refer to our reply to point 6 below for a discussion of the interest in criticality in information propagation on social networks, as well as in other dynamical systems (e.g. brain dynamics, refs [3,8,9]).

2. Results are only for networks without clustering, degree correlations and/or communities. However, it is known that mesoscopic structure like clustering, communities can slow down propagation of cascades (see Karsai, M., Kivela, M., Pan, R. K., Kaski, K., Kertesz, J., Barabasi, A.-L., & Saramaki, J. (2011). Small But Slow World: How Network Topology and Burstiness Slow Down Spreading, 83(2), 025102.) So it might be that long-ranged temporal profile of the cascades is more impacted by the meso and macro structure of the network.

Reply to point 2: This is a very good point. Our mathematical derivation necessarily assumes the absence of reciprocal links, clustering, community structure, degree correlations etc., but we believe that the impact of all these aspects in real networks is superseded by the effect of the degree distribution (which is captured in our theory). Rather than attempting to demonstrate this with further examples of synthetic networks, we instead compare simulation results on an empirical Twitter network with those found on rewired versions of the network, where the rewiring (as in the paper by Karsai et al. that is referenced by the Reviewer, now Ref [43]) is controlled to remove meso- and macro-scopic structure while retaining the effects of degree distribution. The resulting changes and additions (to Figure 8, Supplementary Figure 6 and Supplementary Note 8) are described in detail in our reply to point 5 below.

3. Results are obtained assuming that the temporal dynamics of propagation is Poissonian. However, it is well known that inter-event distributions are not Poissonian and that this yields to a different dynamics. Specifically, non-Poissonian inter-event distributions modify the temporal profile of average branching process so most likely are going to modify that of the cascades (see Iribarren, J. E. L., & Moro, E. (2009). Impact of human activity patterns on the dynamics of information diffusion, 103(3), 038702–038702. <http://doi.org/10.1103/PhysRevLett.103.038702> and [38])

Reply to point 3: Our theoretical derivation indeed assumes Poissonian (Markovian) temporal dynamics. As we note in the Discussion section, extending the theory to non-Markovian cascades is a significant challenge that lies beyond the scope of the current work. However, the Reviewer's comment has inspired us to add Supplementary Note 9, where we use numerical simulations to examine the impact of non-Markovian temporal dynamics on avalanches in the meme propagation model. In this modification of the model of Ref. [32], users become active (send tweets) at times that are separated by intervals τ_n drawn from a unit-mean Weibull distribution with shape parameter k . For $k = 1$, the inter-event time distribution is exponential, and we recover the results of the original Markovian model shown in Fig. 5. In Supplementary Note 9 we successively decrease the parameter k to investigate the impact of increasingly non-Markovian dynamics (see Supplementary Figure 7). Supplementary Figure 8 shows results for $k = 0.5$, which are qualitatively similar to the $k = 1$ Markovian case of Fig. 5, except for some early-time effects (compare, in particular, panels (e) and (f) in Fig. 5 and in Supplementary Fig. 8). Our main result, that the symmetry of the avalanche shape functions depends on whether the offspring distribution (derived from the network degree distribution and the dynamics) has a power-law tail, certainly seems robust to the non-Markovian effects (compare panels (a) and (b)). We continue towards increasingly non-

Markovian dynamics in Supplementary Figs. 9 and 10, where the Weibull shape parameter is, respectively, $k = 0.4$ and $k = 0.3$. The early-time effect is more extreme in these cases, which also impacts upon the early-time avalanche shapes. Nevertheless, we argue that these simulation results indicate that the domain of possible applicability of our theoretical results is not limited solely to Markovian dynamics, and that the current work therefore should be of broad appeal (while stressing that the detailed understanding of non-Markovian effects requires an extensive theoretical analysis, that hopefully builds upon the current work).

We have also added the simulation code for the non-Markovian dynamics to the webpage of codes and datasets that is associated with the paper (<http://www.ul.ie/gleesonj/avalanches>)

4. Results are presented only for the average temporal profile. But how many cascades do we have to observe to reproduce the average temporal profile? As it is shown in figure 1, actual single cascade profiles can be very different from the average temporal profile. By the way, the article does not mention how many cascades are used to obtain the average temporal profiles. This is an important point, since it seems that from equation (10), the coefficient of variation diverges as $1/\sqrt{t}$ for small t (in the case of scale-free networks).

Reply to point 4: We point out that the information on the number of cascades used to obtain the average temporal profiles is already contained in the Methods section, under “Details of numerical simulations”, where values are given for n_A for each of Figures 5 through 8. Inspired by the Reviewer’s comment, however, we have also added three new Supplementary Figures and Supplementary Note 7 to give examples of how changing the number of avalanches n_A affects the results. Taking as a baseline the neuronal dynamics case of Fig. 6, which has $n_A = 10^7$, we show the corresponding results that are obtained when lower values of n_A are used, reducing the number of avalanches by an order of magnitude for each figure: $n_A = 10^6$ in Supplementary Figure 3, $n_A = 10^5$ in Supplementary Figure 4, and $n_A = 10^4$ in Supplementary Figure 5. As discussed in Supplementary Note 7, the qualitative results remain robust to these changes, albeit with increasingly large error bars. In particular we use these results to highlight how the non-terminating avalanche shape (panels (c)) can give useful information from smaller datasets that cause the traditional avalanche shape diagnostic to fail due to lack of data: in Supplementary Fig. 5(a), for example, there are no avalanches that terminate at precisely $T=40$ (and so the black curve is absent), but the corresponding curve for the non-terminating avalanches at $T=40$ (panel (c)) is still well-defined. We have added text to highlight why this may be useful to experimentalists who are seeking to identify whether a given dynamics is critical or not.

5. Results are presented mainly for synthetic networks and only a final comparison of the meme propagation model on one instance of empirical Twitter network is done. Although the authors claim their results justify the predictions of the theory, one might wonder what is the statistical significance of this claim. That is, if we are only given the information displayed in figure 8b, can one establish that the model is mainly explaining what we are observing?

Reply to point 5: This is a valid point, and to address it, we have decided to completely change Figure 8. It now includes panels showing the non-terminating avalanche shapes and the average number of tweeting events (panels (c) and (e) of Fig. 8). In addition (and to also answer point 2 above), we compare the simulation results on the empirical Twitter network with those conducted on a suitably rewired version of the original network, in order to check whether the non-symmetric avalanche shapes we see are due mainly to the degree distribution of the network (as argued in our theory), or

possibly affected by other characteristics present in the empirical network, such as the high fraction of reciprocal links (which is 48%), correlations, communities, etc.

For the panels in the right-hand column of Fig. 8 (panels (b), (d) and (f)), we use a directed Erdos-Renyi network with the same mean degree as the empirical network but with all other structural characteristics randomised. We create this network by simply taking each directed link of the original network and reassigning both its end nodes to be randomly-chosen nodes. This rewired network is used in Fig. 8 to show that symmetric avalanche shapes are obtained when the degree distribution does not have fat tails, as expected from our theory.

In Supplementary Note 8 we describe a second rewiring algorithm, showing the corresponding results in Supplementary Fig. 6. This rewiring algorithm (called “ p_{jk} -rewiring” in Ref. [32]) preserves the degree distribution of the original network, but removes (or drastically reduces) clustering, reciprocal links, and any macro-scale structure (similar to the rewiring algorithm used in Karsai et al., now Ref. [43]). The algorithm begins by severing all links of the original network, but with each node retaining its number j of “in-stubs” and k of “out-stubs” that represent the endpoints of the deleted original edges. We then randomly select one out-stub and one in-stub from the entire set of stubs and create a new edge joining the selected nodes. By repeating this procedure, each time selecting from the set of unused out-stubs and in-stubs, we create a network with precisely the same (j, k) distribution as the original network, but with randomised meso- and macro-scopic structure: for example, the fraction of reciprocal links is reduced from 48% in the original network to 1% in the rewired network. In Supplementary Fig. 6 we show simulation results for this rewired network, for comparison with those on the original network in panels (a), (c) and (e) of Fig. 8. The asymmetric avalanche shape remains clearly evident in the rewired case, which is evidence that the fundamental driver of the avalanche shape asymmetry is indeed the degree distribution (as examined in our theory), while meso- and macro-scopic structural effects are of lesser importance.

6. In this regard, the paper lacks an important point for the community: what is the applicability of the results presented? Do the authors already know of or have in mind any domain in which those results can have an important impact? The spreading of information cascades is a highly active field (the authors have work on that) and many datasets have been collected of hashtags spreading in Twitter, links in blogs, news in Facebook, etc. Some of those datasets are available and the authors could have applied their theory to see if it applies to any experimental result.

Reply to point 6: The Reviewer raises an excellent point: the application of these results to information cascades, and to other dynamics where critical cascades are expected, is the main motivation for this study. We expect two broad areas of application. First, as mentioned by Reviewer #1, the growing number of experimentalists who use temporal avalanche shape analysis to detect criticality in dynamics (e.g., in the human brain or in ferromagnetic materials, Refs [1-4]), will find our discussion of why asymmetric avalanche shape arise helpful, and (probably more importantly) use our proposed new measures of non-terminating avalanche shapes and average number of events (see panels (c) through (f) of Figs. 5-8) to detect criticality in experiments. The non-terminating avalanche shapes are likely to prove important in cases where data is difficult to obtain, as they require fewer avalanches to construct than the corresponding average avalanche shape (see the discussion in reply 4 above).

Secondly, we anticipate that the analysis we use here will indeed be applicable to information cascades, as suggested by Reviewer #2; in particular to test the hypothesis that the dynamics of spreading of information over social networks may, in some cases, be poised near a critical point (e.g., Ref [32]). However, as the Reviewer has correctly noted in point 3 above, the dynamics of

social systems are highly non-Markovian, and so a rigorous analysis of data from such systems requires the development of a theory for non-Markovian avalanches, which is beyond the scope of the current paper (although, as noted in reply 3, we believe this paper will be a foundation stone of such a theory). As an example of the type of application that we believe should be possible to rigorously analyse in the future, we show here the calculation of the average number of tweets per day at a time t after the avalanche starts (the same plot as used in panels (e) and (f) of Figs. 5-8) for the empirical data on the popularity of 1.4×10^5 hashtags

associated with the Spanish 15M protest that is used in Ref. [34]. The strongly non-Markovian nature of the dynamics is evidenced by the high number of average events early in the cascade (especially in day $t = 1$), followed by a long period of roughly constant number of tweets per day (punctuated by smaller spikes that are probably due to the occurrence of exogeneous events during the 15M movement, see González-Bailón et al., Sci. Rep. 1:197 (2011)). The similarity to, for example, panel (f) of Supplementary Figure 10 is striking, and we anticipate that further work (using this paper as a theoretical foundation) will use such methods to rigorously examine criticality in information spreading, as well as in other domains of application (e.g., neural avalanches, Barkhausen noise, etc.).

We have added text to the abstract and Discussion to clarify the applicability of the results as outlined here.

Given the points above, I found the results presented very interesting, but perhaps only for a specialized community working on theoretical aspects of cascades dynamics.

Reply to Reviewer #2: We thank the Reviewer for his/her positive opinion of the theoretical results. We hope that our replies, and the corresponding material that we have added in both the paper and the Supplementary Notes, will convince him/her that the theory, despite the restrictions required for mathematical proof, can nevertheless be applied to help understand important qualitative features of cascades that are of a broad interest to the readership.

REVIEWERS' COMMENTS:

Reviewer #1 (Remarks to the Author):

The authors have satisfactorily addressed my comments.